# BIN overlap confirms transcontinental distribution of pest aphids (Hemiptera: Aphididae)

**Muhammad Tayyib Naseem**[1,2], **Muhammad Ashfaq**[3]*, **Arif Muhammad Khan**[1,4], **Akhtar Rasool**[1,5], **Muhammad Asif**[1], **Paul D. N. Hebert**[3]

**1** National institute for Biotechnology and Genetic Engineering, Faisalabad, Pakistan, **2** Pakistan Institute of Engineering and Applied Sciences, Islamabad, Pakistan, **3** Centre for Biodiversity Genomics & Department of Integrative Biology, University of Guelph, Guelph, ON, Canada, **4** Department of Biotechnology, University of Sargodha, Sargodha, Pakistan, **5** Department of Zoology, University of Swat, Swat, Pakistan

* mashfaq@uoguelph.ca

## Abstract

DNA barcoding is highly effective for identifying specimens once a reference sequence library is available for the species assemblage targeted for analysis. Despite the great need for an improved capacity to identify the insect pests of crops, the use of DNA barcoding is constrained by the lack of a well-parameterized reference library. The current study begins to address this limitation by developing a DNA barcode reference library for the pest aphids of Pakistan. It also examines the affinities of these species with conspecific populations from other geographic regions based on both conventional taxonomy and Barcode Index Numbers (BINs). A total of 809 aphids were collected from a range of plant species at sites across Pakistan. Morphological study and DNA barcoding allowed 774 specimens to be identified to one of 42 species while the others were placed to a genus or subfamily. Sequences obtained from these specimens were assigned to 52 BINs whose monophyly were supported by neighbor-joining (NJ) clustering and Bayesian inference. The 42 species were assigned to 41 BINs with 38 showing BIN concordance. These species were represented on BOLD by 7,870 records from 69 countries. Combining these records with those from Pakistan produced 60 BINs with 12 species showing a BIN split and three a BIN merger. Geo-distance correlations showed that intraspecific divergence values for 49% of the species were not affected by the distance between populations. Forty four of the 52 BINs from Pakistan had counterparts in 73 countries across six continents, documenting the broad distributions of pest aphids.

## Introduction

Although aphids (Hemiptera: Aphididae) are important plant pests, their life stage diversity and phenotypic plasticity have constrained the development of effective taxonomic keys in the past [1,2]. With over 5,000 described species, the Aphididae represents the largest family within the Aphidoidea [3]. Most pest aphids belong to the subtribe Aphidina which includes

**Data Availability Statement:** The specimen metadata and sequences generated in this study are available on the Barcode of Life Data System

(www.boldsystems.org (BOLD) in the dataset DS-MAAPH.

**Funding:** This research was supported by a grant from the Higher Education Commission (HEC), Pakistan and by grant 106106-001 (Engaging Developing Nations in iBOL) from IDRC. MT Naseem was supported by the Higher Education Commission (HEC), Pakistan under the Indigenous PhD Fellowship. This is a contribution to the Food from Thought project supported by the Canada First Research Excellence Fund.

**Competing interests:** The authors have declared that no competing interests exist.

670 described species so far [3,4]. Nearly 100 aphid species have been listed as serious agricultural pests; they damage more than 300 plant species [5,6], and lower crop yield by direct feeding and by transmitting viral diseases [7].

Sibling species complexes are common in many pest aphids [8]. Very often, these species are morphologically identical but genetically distinct [9]. They often include anholocyclic biotypes (= clones) with differing host preferences and varying competency for disease transmission [10,11]. Species identification is so challenging that taxonomic keys are either ineffective or only useful for a particular geographic area or taxonomic group [12]. These deficits have prompted the search for alternative approaches for identification such as protein profiling [13] and the use of DNA sequence data [14,15]. However, the later approach has gained stronger uptake due to its universal applicability, low cost, and strong performance [16].

Past studies have demonstrated that DNA-based approaches can enable both specimen identification and the clarification of putative cryptic species complexes [17,18]. Diverse mitochondrial and nuclear genes have been used individually and in combination to discriminate insect species [19–21]. Although multigene analyses are valuable in resolving complex taxonomic situations and essential for phylogenetic reconstructions [22,23], it has seen little application in routine identifications [18]. By contrast, DNA barcoding [24] employs a 658 base pair (bp) segment of a single mitochondrial gene, cytochrome *c* oxidase I, to discriminate animal species. Because of its ease of application, DNA barcoding has become the most popular approach for the identification of specimens in diverse insect groups including aphids [25–32]. Its effortless integration with high-throughput sequencing workflows has made DNA barcoding an effective tool for large-scale pest diagnosis, biosurveillance, and biodiversity assessments [33–35].

The application of DNA barcoding requires bioinformatics support and a comprehensive reference library [36]. The Barcode of Life Data System (BOLD– www.boldsystems.org) [37] meets the former need and currently includes more than six million barcode records from animals. Most of these records are from insects (5.2 million) and 49,000 of them derive from aphids (accessed 3 July 2019). All barcode sequences meeting quality criteria receive a Barcode Index Number (BIN) [38]. BINs are an effective species proxy because they correspond closely with species designated through morphological study [39,40]. As a result, BINs are now routinely employed for biodiversity assessments, counting species, analyzing cryptic species complexes, and assessing species ranges [41–43]. All these developments have generated considerable interest in DNA barcoding, leading to the development of well-parameterized reference barcode libraries for some groups at continental and global scales [32,44–48].

Although the DNA barcode library for insects is still incomplete, it is already highly valuable for identifying various pest species and assessing their distributions [29,45,49–52]. However, the lack of reference sequences constrains the utility of DNA barcoding in many situations. Although barcode coverage for the aphid fauna of some countries is extensive [46,53,54], DNA barcoding studies in other nations, including Pakistan, for these pest species are limited. The current study addresses this gap by generating a barcode reference library for the pest aphids of Pakistan, and by using BINs to reveal their links to aphid assemblages in other regions.

## Materials and methods

### Ethics statement

No specific permissions were required for this study. The study did not involve endangered or protected species.

Aphids were sampled from 123 plant species representing 33 families at 87 sites in Pakistan (Fig 1, S1 Table) during 2010–2013. These sites included agricultural settings, nurseries, national parks, botanical gardens, natural forests, and disturbed habitats. Based on GPS

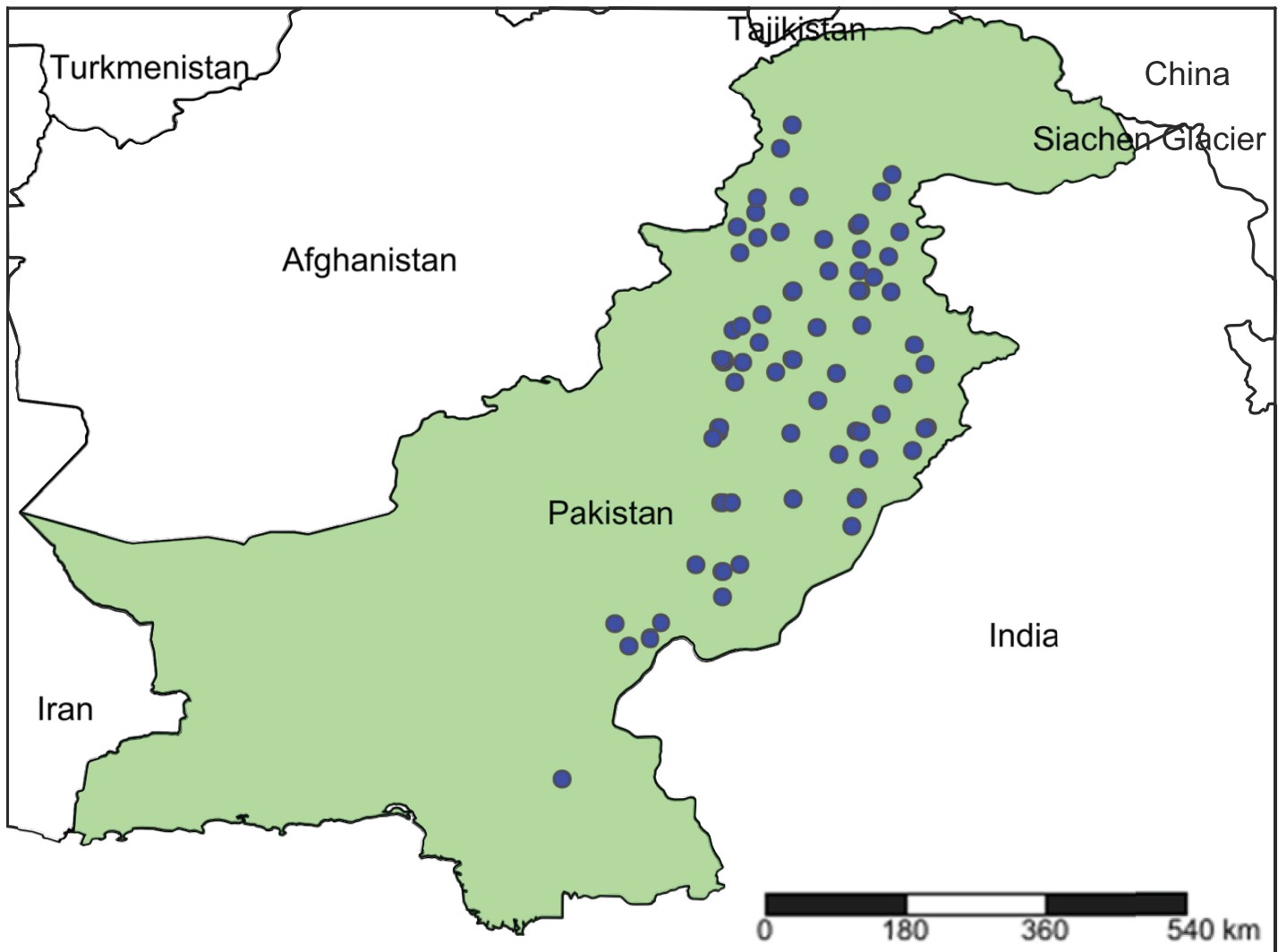

**Fig 1. Collection sites for aphids in Pakistan.** The map was generated by www.simplemappr.net using GPS coordinates.

coordinates, the collection sites were rendered using SimpleMappr.net (Fig 1). Aphids were collected by either beating foliage above a white paper sheet or by removing them from their host plant with a camel hair brush [55]. Collections were transferred into Eppendorf tubes prefilled with 95% ethanol and stored at -20˚C until analysis.

### Identification

Aphids were identified using standard taxonomic keys [55,56]. Morphological characters were examined with a Labomed CZM6 stereomicroscope (Labo America) equipped with an ocular micrometer. Each specimen was identified to species-level based on morphology. This identification was later validated by DNA barcode sequence matches on BOLD.

### DNA barcoding

Front-end processing, including specimen sorting, arraying, databasing, and imaging was performed at the Insect Molecular Biology Laboratory, National Institute for Biotechnology and

Genetic Engineering (NIBGE), Faisalabad. Individual specimens were placed into 96-well format in a microplate pre-filled with 30 μl of 95% ethanol in each well. Each specimen was photographed dorsally using a 12 megapixel Olympus μ-9000 camera (Olympus America Inc., USA) mounted on a stereomicroscope. Specimen metadata (collection information and taxonomy) and images were submitted to BOLD under the project MAAPH, "Barcoding Aphid Species of Pakistan". DNA extraction, PCR amplification, and sequencing were carried out at Centre for Biodiversity Genomics at Guelph. DNA extraction followed Ivanova et al. [57] with voucher recovery protocol [58]. PCR amplification of the COI-5′ (barcode region) [24] was performed using primer pair C_LepFolF (forward) and C_LepFolR (reverse) (http://ccdb.ca/site/wp-content/uploads/2016/09/CCDB_PrimerSets.pdf) in 12.5 μL reactions that included standard PCR ingredients [59] and 2 μL of DNA template. The thermocycling regime was: 94˚C (1 min), 5 cycles at 94˚C (40 s), 45˚C (40 s), 72˚C (1 min); 35 cycles at 94˚C (40 s), 51˚C (40 s), 72˚C (1 min); and a final extension at 72˚C (5 min). PCR success was verified by analyzing the amplicons on 2% agarose E-gel® 96 system (Invitrogen Inc.). Specimens which failed to amplify in the first round of PCR were re-run with primers LepF2_t1 (TGTAAAACGACG GCCAGTAATCATAARGATATYGG) [60] and LepR1 using the same PCR conditions. PCR products were sequenced bidirectionally on an Applied Biosystems 3730XL DNA Analyzer using the BigDye Terminator Cycle Sequencing Kit (v3.1) (Applied Biosystems). Sequences were edited using CodonCode Aligner (CodonCode Corporation, USA), and translated on MEGA v6 [61] to confirm they were free of stop codons, and submitted to BOLD. The specimen metadata and sequences generated in this study are available on BOLD in the dataset DS-MAAPH. Vouchers were deposited at the Insect Museum, NIBGE, Faisalabad, Pakistan (with sample ID prefix NIBGE) and at the Centre for Biodiversity Genomics, Guelph, ON, Canada (with ID prefix BIOUG).

## Data analysis

All barcode sequences were compared with those on BOLD and GenBank to ascertain sequence similarities. Sequence matches on BOLD were obtained using the "Identification Engine" tool whereas nBLAST (http://www.ncbi.nlm.nih.gov/blast/) was used on GenBank. All sequences meeting quality standards (>500 bp, <1% ambiguous bases, no stop codon or contamination flag) were assigned to a BIN [38] (as of 18 January 2019). BIN discordance and BIN overlap reports were generated using analytical tools on the BOLD workbench. As a test of the reliability of species discrimination, the presence or absence of a 'barcode gap' [62] among species was determined on BOLD. A species was considered distinct when its maximum intraspecific distance was less than the distance to its nearest neighbor (NN).

ClustalW nucleotide alignments [63] and neighbor-joining (NJ) analysis [63] were conducted in MEGA6. The NJ analysis employed the Kimura-2-Parameter (K2P) [64] distance model, with pairwise deletion of missing sites, and 1000 non-parametric bootstrap [65] replicates for the nodal support. Bayesian inference was performed in MrBayes v3.2.0 [66] employing the Markov Chain Monte Carlo (MCMC) technique. This analysis was based on representative sequences from 67 aphid haplotypes in the dataset extracted using DNaSP v5.10 [67] with *Diaphorina citri* (Hemiptera: Psyllidae) as outgroup. The data were partitioned in two ways; i) a single partition with parameters estimated across all codon positions, ii) a codon-partition in which each codon position was allowed different parameter estimates. The evolution of sequences was modelled by the GTR+Γ model independently for the two partitions using the "unlink" command in MrBayes, and the best model was selected using FindModel (www.hiv.lanl.gov/cgi-bin/findmodel/findmodel.cgi). The analyses were run for 10 million generations using four chains with sampling every 1000 generations. Bayesian

posterior probabilities were calculated from the sample points once the MCMC algorithm converged. Convergence was defined as the point where the standard deviation of split frequencies was less than 0.002 and the PSRF (potential scale reduction factor) approached 1, and both runs converged to a stationary distribution after the burn-in (by default, the first 25% of samples were discarded). Each run produced 100001 samples of which 75001 samples were included. The trees generated through this process were visualized using FigTree v1.4.0.

BOLD was searched for barcode records for the 42 species encountered in this study. The resultant records were examined for BIN assignment [38] and used in a geo-distance correlation analysis to examine the relationship between geographic and genetic distance in each species. Two methods were employed in the latter analysis; the Mantel Test [68] was used to examine the relationship between geographic (km) and genetic (K2P) distance matrices, while the second approach compared the spread of the minimum spanning tree of collection sites and maximum intra-specific divergence [69]. The relationship between geographic and intra-specific distances was analyzed for each species with at least one individual from three or more sites. BINs recovered from Pakistan were also used in BIN-overlap analysis on BOLD to ascertain the incidence of unique BINs in a region, and to estimate overlap in BIN composition.

## Results

Morphological analysis facilitated by the barcode data enabled identification of most specimens (774/809) and revealed 42 species, each representing an important crop pest (S1 Table). Another 32 specimens could be placed to a genus while the remaining three could only be assigned to a subfamily (Aphidinae). Overall, the 809 specimens included representatives of 30 genera and five subfamilies (Aphidinae, Calaphidinae, Chaitophorinae, Eriosomatinae, Lachninae) of the Aphididae (S2 Table). Members of the Aphidinae were dominant (n = 780) as the other four subfamilies were represented by just 29 specimens with Chaitophorinae and Lachninae each contributing one specimen (S2 Table). Among the determined genera, *Aphis* was most common one (n = 306), and was represented by eight identified and three undetermined species. Furthermore, *Myzus* was the second most frequent genus (n = 170), but it was only represented by one species, *Myzus persicae*. *Rhopalosiphum*, the third most abundant (83) genus, was represented by three major pest species (*R. maidis*, *R. padi*, *R. rufiabdominale*). Two species (*Aphis astragalina*, *Periphyllus lyropictus*) represented first records for Pakistan whereas two others (*Lipaphis pseudobrassicae*, *Sarucallis kahawaluokalani*) were known, but were recorded as *Lipaphis erysimi* and *Tinocallis kahawaluokalani*.

The 809 barcode sequences provided two or more records for 36 of the 42 species and single records for the rest (Tables 1 and S1). Maximum K2P divergence values within species ranged from 0–3.6% (mean = 0.1%), whereas distance values within genera were between 0.8–10.3% (mean = 7.4%), and within family (Aphididae) 3.7–17.3% (mean = 9.6%) (Table 1). Barcode gap analysis examined the ability of barcodes to discriminate the 42 named species. With the exception of one species (*Aphis gossypii*), where the maximum intraspecific distance (3.6%) overlapped with *A. affinis*, the maximum intraspecific distance for each species was less than

**Table 1. Sequence divergence (K2P) in the COI barcode region for aphid species from Pakistan with more than three specimens, genera with three or more species, and families with three or more genera.** This analysis only considers specimens that were assigned to a Linnaean species.

| Distance class | n | Taxa | Comparisons | Min (%) | Mean (%) | Max (%) |
|---|---|---|---|---|---|---|
| Within Species | 764 | 36 | 30756 | 0 | 0.1 | 3.6 |
| Within Genus | 434 | 6 | 32305 | 0.8 | 7.4 | 10.3 |
| Within Family | 770 | 1 | 233004 | 3.7 | 9.6 | 17.3 |

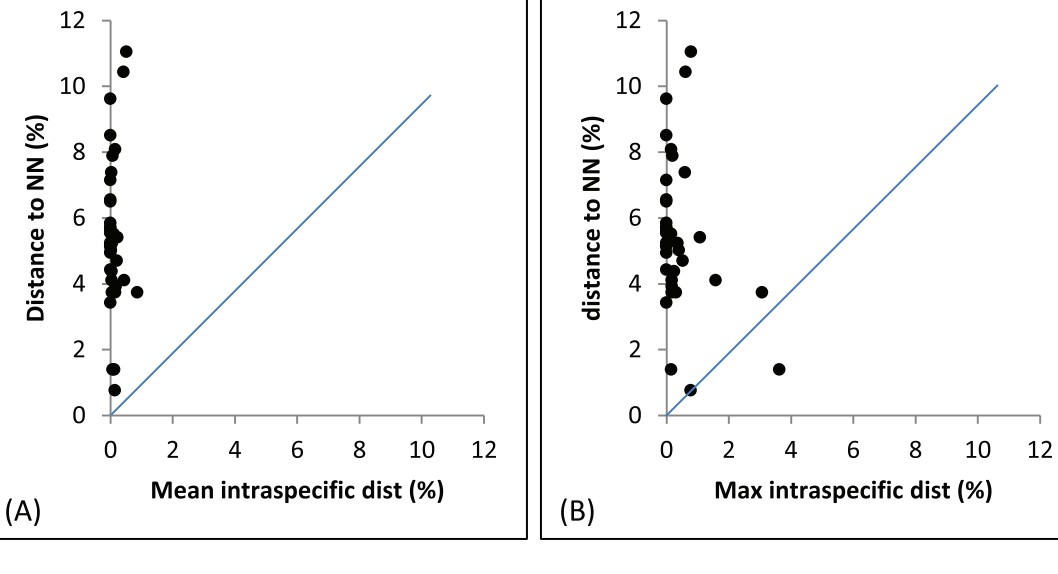

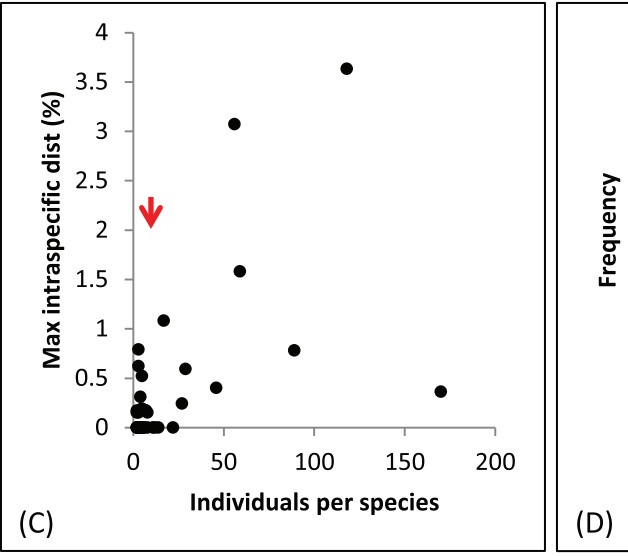

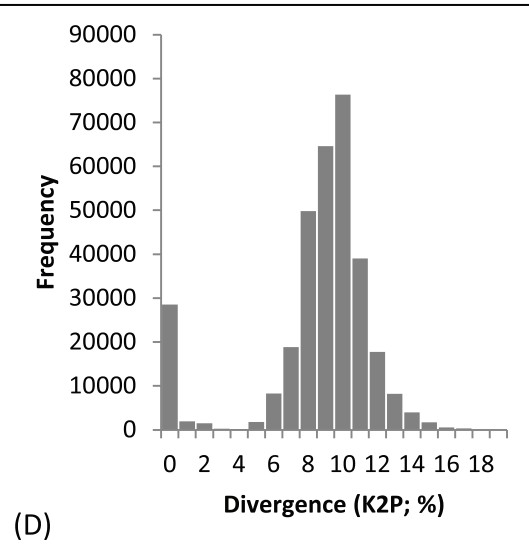

**Fig 2.** Barcode gap analysis for species of aphids with three or more specimens collected in Pakistan. NN = nearest neighbor.

its NN distance (Fig 2A, Fig 2B). This pattern did not change with increased sample size (Fig 2C).

Nearly all sequences (801/809) fulfilled the criteria for a BIN assignment, and they were placed in 52 BINs. The 774 specimens of the 42 species were assigned to 41 BINs; 38 showed BIN concordance (species members = BIN members), one species (*Rhopalosiphum padi*) was split (AAA9899, ACF2924), and two species (*Aphis affinis*, *A. gossypii*) were merged (AAA3070), while another, *Aphis astragalina* lacked a BIN assignment due to its low quality sequence (410 bp, 9 Ns). The 32 specimens lacking a species assignment were placed in 9 BINs–three for *Aphis* and one for each of the other six genera (*Acyrthosiphon*, *Capitophorus*, *Forda*, *Hyalopterus*, *Macrosiphoniella*, *Schizaphis*). The three specimens only identified to a subfamily were assigned to two BINs. NJ analysis (Fig 3) and Bayesian inference (BI) (Fig 4) supported the monophyly of each of the 52 BINs. The NJ and BI also discriminated the species

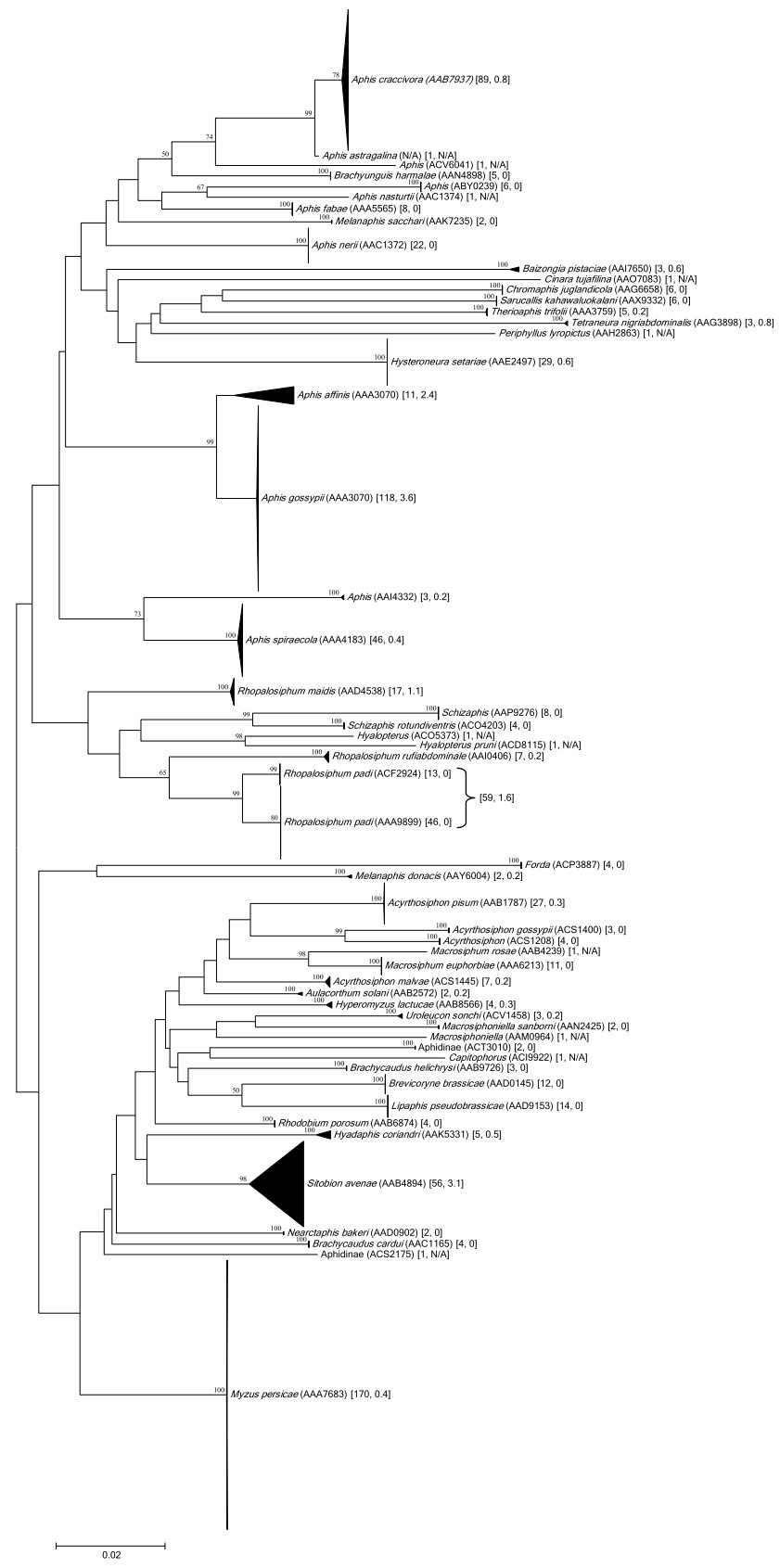

**Fig 3.  NJ analysis of COI-5′ sequences from species/BINs of aphids from Pakistan.** Bootstrap values (%) (1,000 replicates) are shown above the branches (values <50% are not shown) while the scale bar shows K2P distances. The node for each species/BIN with multiple specimens was collapsed to a vertical line or triangle, with the horizontal depth indicating the level of intraspecific divergence. BIN numbers are shown for species with only family- or genus-level identification or those split into two BINs.

or genera that either lacked (*Aphis astragalina*) or shared BINs (*Aphis gossypii*, *A. affinis*), as they formed distinct branches on the NJ and BI trees (Fig 3, Fig 4).

Geo-distance correlation analysis for 37 species was conducted by including an additional 5,067 sequences from conspecific individuals deposited on BOLD. This analysis showed that intraspecific divergences in 49% of the species were not affected by expanding analysis to consider their entire ranges (Mantel test, P>0.01) (Table 2). The other 51%, that were affected by geographic range, included six species with BIN splits and eight with intraspecific divergence higher than >2%. The distributional patterns of aphids detected in Pakistan were further analyzed by examining BIN overlap between Pakistan and other countries, a comparison that involved 9,905 barcode records assigned to the 52 BINs. This analysis showed that 27 of the 52 BINs were recorded from four or more continents while eight were unique to Pakistan (Table 3). Except for *Acyrthosiphon malvae* and *Uroleucon sonchi*, all named species (40) analyzed in this study already had barcode records from multiple countries and continents (Table 3).

## Discussion

Prior morphological surveys on the aphids of Pakistan have reported the presence of nearly 300 species [70–72]. Most of this work focused on specific geographic regions [73] or species attacking crops [74,75]. The current study surveyed aphids across major agricultural areas of Pakistan from a wider range of host plants, but primarily aimed to develop a barcode reference library for the fauna for the first time. Prior studies have begun to construct barcode reference libraries for some pest insect groups, such as aphids in Canada [29], leafminers in USA [76], fruit flies in Africa [45], food pests in Korea [49], thrips in Pakistan [28], looper moths in British Columbia [77], and mealybugs in China [52]. These libraries have stimulated the use of DNA barcoding in biosecurity and plant protection programs [78], but their use revealed the need for expanded parameterization of the libraries in order to improve their utility in diagnosing newly encountered species. Barcode libraries for two major pest insect groups in Pakistan, thrips and whiteflies, have progressed well [28,79], but other groups have seen little attention in this country so far. The current study not only expands on the prior efforts by barcoding another group of insect pests but also maps the global presence of pest aphids by using BINs.

Most aphids analyzed in this study were assigned to a species, but 35 specimens could only be determined to genus or subfamily level. In part, this difficulty reflected the fact that many important pest aphids are cryptic species complexes whose members are almost impossible to discriminate using morphological traits only [42] or their identification was beyond our expertise. For example, *Aphis gossypii* is a particularly challenging species complex [5,13]; it includes at least 18 morphologically indistinguishable species [80] likely explaining its wide range of primary and secondary host plants [81]. In the present study, DNA barcoding separated all eight species of the genus *Aphis* that were encountered. Although K2P distances between two species pairs; i) *A. affinis* and *A. gossypii* (1.4%), ii) *A. astragalina* and *A. craccivora* (0.8%) were low, both NJ analysis and Bayesian inference supported the monophyly of each species. The COI divergences in this study are similar to those reported in prior investigations

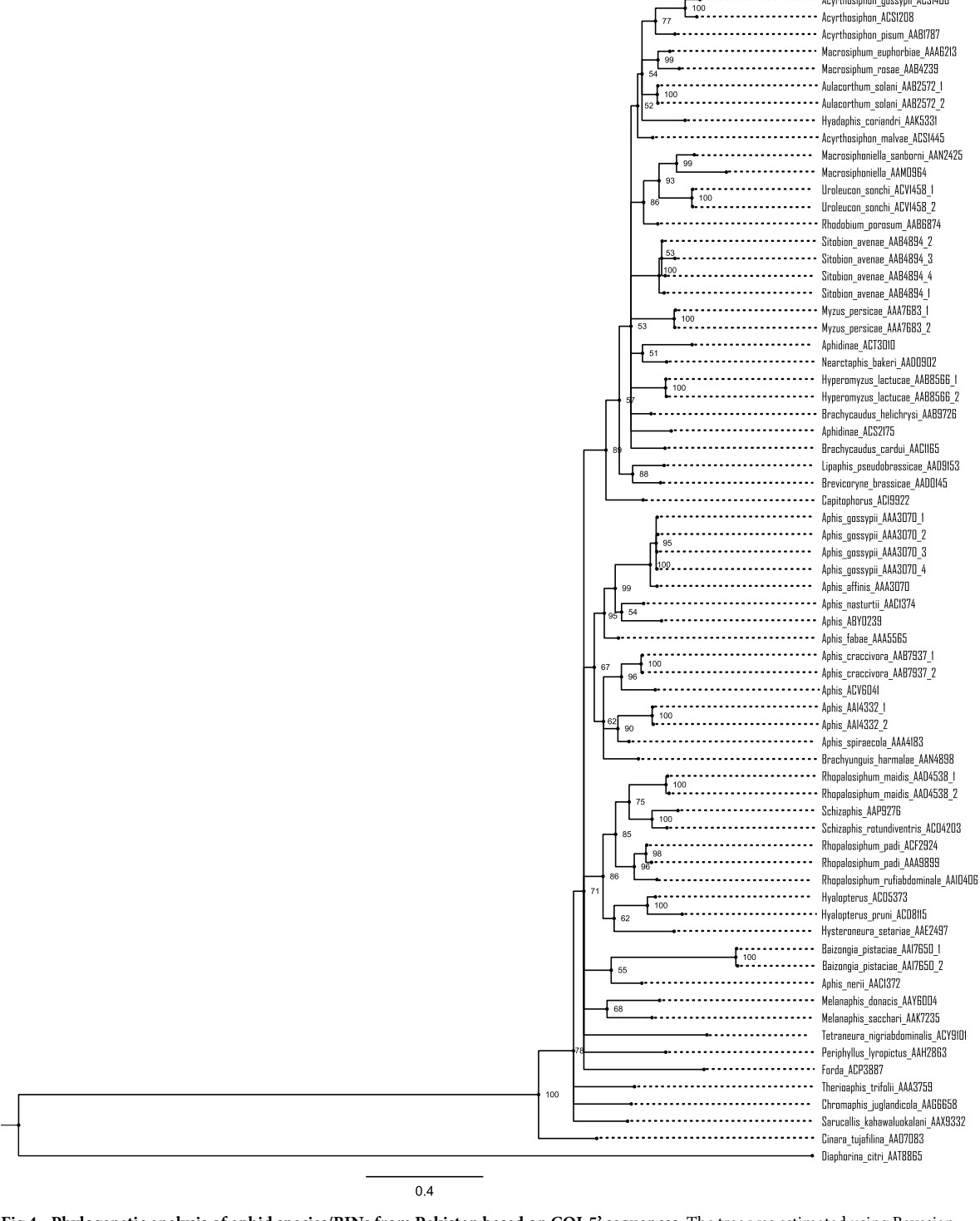

**Fig 4. Phylogenetic analysis of aphid species/BINs from Pakistan based on COI-5' sequences.** The tree was estimated using Bayesian inference. Posterior probabilities are indicated at nodes. The analysis was based on representative sequences from 67 aphid haplotypes in the dataset that were extracted using DnaSP v5.10 (Librado and Rozas 2009). Taxa are followed by the BINs and haplotype numbers. *Diaphorina citri* (BOLD:AAT8865) was employed as outgroup.

**Table 2. Geographic (km) and genetic (K2P) distance correlation analysis for 42 aphid species from Pakistan combined with conspecifics from 69 other countries.**

| Species | Record Count | BINs | Linear Regression $R^2$ | Gen Dist Max | Geo Dist Max | Mantel $R^2$ | Mantel $P$ value |
|---|---|---|---|---|---|---|---|
| *Acyrthosiphon malvae* | 54 | 2 | 0.13 | 4.8 | 18675 | 0.13 | 0.01 |
| *Acyrthosiphon pisum* | 205 | 1 | 0.01 | 1.7 | 19312 | 0.01 | 0.05 |
| *Aphis affinis* | 8 | 1 | 0.18 | 0.2 | 356 | 0.18 | 0.04 |
| *Aphis astragalina* | 37 | 1 | 0.71 | 0.8 | 10789 | 0.71 | 0.01 |
| *Aphis craccivora* | 420 | 1 | 0.09 | 1.4 | 19417 | 0.00 | 0.01 |
| *Aphis fabae* | 426 | 1 | 0.01 | 2.5 | 19456 | 0.01 | 0.01 |
| *Aphis gossypii* | 362 | 1 | 0.00 | 3.9 | 19369 | 0.00 | 0.8 |
| *Aphis nasturtii* | 38 | 1 | 0.48 | 1.6 | 11656 | 0.48 | 0.01 |
| *Aphis nerii* | 99 | 1 | 0.02 | 1.2 | 19110 | 0.02 | 0.01 |
| *Aphis spiraecola* | 277 | 2 | 0.00 | 3.1 | 19355 | 0.00 | 0.2 |
| *Aulacorthum solani* | 118 | 1 | 0.13 | 2.0 | 19291 | 0.13 | 0.01 |
| *Baizongia pistaciae* | 5 | 2 | 0.48 | 4.4 | 5967 | 0.48 | 0.23 |
| *Brachycaudus cardui* | 54 | 1 | 0.17 | 0.9 | 11929 | 0.17 | 0.01 |
| *Brachycaudus helichrysi* | 108 | 4 | 0.01 | 3.1 | 19426 | 0.01 | 0.01 |
| *Brevicoryne brassicae* | 166 | 2 | 0.02 | 6.3 | 19178 | 0.02 | 0.04 |
| *Chromaphis juglandicola* | 11 | 1 | 0.52 | 0.8 | 10908 | 0.52 | 0.01 |
| *Hyadaphis coriandri* | 13 | 1 | 0.40 | 1.4 | 685 | 0.40 | 0.01 |
| *Hyalopterus pruni* | 151 | 3 | 0.06 | 6.6 | 16867 | 0.06 | 0.01 |
| *Hyperomyzus lactucae* | 87 | 1 | 0.05 | 0.3 | 19474 | 0.05 | 0.02 |
| *Hysteroneura setariae* | 53 | 1 | 0.04 | 1.9 | 15843 | 0.04 | 0.01 |
| *Lipaphis pseudobrassicae* | 91 | 2 | 0.12 | 3.6 | 18212 | 0.13 | 0.01 |
| *Macrosiphoniella sanborni* | 5 | 1 | 0.01 | 0.2 | 15935 | 0.01 | 0.43 |
| *Macrosiphum euphorbiae* | 198 | 1 | 0.14 | 1.6 | 19482 | 0.14 | 0.01 |
| *Macrosiphum rosae* | 80 | 1 | 0.00 | 1.2 | 19379 | 0.00 | 0.14 |
| *Melanaphis donacis* | 7 | 1 | 0.16 | 0.2 | 6092 | 0.16 | 0.35 |
| *Melanaphis sacchari* | 225 | 1 | 0.02 | 1.9 | 18400 | 0.02 | 0.01 |
| *Myzus persicae* | 322 | 1 | 0.00 | 2.2 | 19234 | 0.00 | 0.68 |
| *Nearctaphis bakeri* | 70 | 2 | 0.04 | 6.8 | 11961 | 0.04 | 0.09 |
| *Periphyllus lyropictus* | 33 | 1 | 0.00 | 0.2 | 11002 | 0.00 | 0.92 |
| *Rhodobium porosum* | 7 | 1 | 0.08 | 0.2 | 12350 | 0.08 | 0.1 |
| *Rhopalosiphum maidis* | 63 | 1 | 0.00 | 2.0 | 18405 | 0.00 | 0.40 |
| *Rhopalosiphum padi* | 1189 | 2 | 0.30 | 5.0 | 19841 | 0.29 | 0.01 |
| *Rhopalosiphum rufiabdominale* | 18 | 1 | 0.03 | 0.2 | 15787 | 0.03 | 0.99 |
| *Sitobion avenae* | 314 | 1 | 0.02 | 4.6 | 18405 | 0.00 | 0.01 |
| *Tetraneura nigriabdominalis* | 41 | 2 | 0.16 | 8.6 | 16534 | 0.16 | 0.01 |
| *Therioaphis trifolii* | 470 | 2 | 0.00 | 13.0 | 18823 | 0.00 | 0.33 |
| *Uroleucon sonchi* | 45 | 2 | 0.11 | 2.3 | 19003 | 0.11 | 0.03 |
| *Acyrthosiphon gossypii* | 5 | 1 | N/A | N/A | N/A | N/A | N/A |
| *Brachyunguis harmalae* | 8 | 1 | N/A | N/A | N/A | N/A | N/A |
| *Cinara tujafilina* | 8 | 1 | N/A | N/A | N/A | N/A | N/A |
| *Sarucallis kahawaluokalani* | 10 | 1 | N/A | N/A | N/A | N/A | N/A |
| *Schizaphis rotundiventris* | 5 | 1 | N/A | N/A | N/A | N/A | N/A |

N/A: Data for the correlation analysis was missing.

[29,82,83] which reported low sequence divergence between sibling species such as *A. gossypii* and *A. affinis* [29].

**Table 3. Occurrence of 52 pest aphid BINs across six continents and their association with Linnaean species on the Barcode of Life Data System (BOLD).**

| BIN | Countries | Continents | (Number) and names of the associated species |
|---|---|---|---|
| AAA3070 | 44 | 6 | (35) *Aphis affinis, A. aliena, A. argrimoniae, A. cf. frangulae, A. chloris, A. cisticola, A. clerodendri, A. confusa, A. crepidis, A. egomae, A. frangulae, A. gossypii, A. hieracii, A. hypericiphaga, A. hypochoeridis, A. idaei, A. leontodontis, A. lichtensteini, A. longirostrata, A. madderae, A. mamonthovae, A. monardae, A. nivalis, A. oestlundi, A. origani, A. parietariae, A. punicae, A. ruborum, A. sedi, A. serpylli, A. sumire, A. taraxacicola, A. teucrii, A. viticis* |
| AAA3759 | 15 | 6 | (1) *Therioaphis trifolii* |
| AAA4183 | 29 | 6 | (1) *Aphis spiraecola* |
| AAA5565 | 31 | 6 | (9) *Aphis fabae, A. solanella, A. hederae, A. ilicis, A. viburni, A. newtoni, A. fukii, A. lambersi, A. seselii* |
| AAA6213 | 19 | 6 | (13) *Macrosiphum albifrons, M. cerinthiacum, M. cholodkovskyi, M. corydalis, M. daphnidis, M. euphorbiae, M. gaurae, M. gei, M. hellebori, M. impatientis, M. sileneum, M. valerianae, M. zionense* |
| AAA7683 | 22 | 6 | (1) *Myzus persicae* |
| AAA9899 | 16 | 4 | (1) *Rhopalosiphum padi* |
| AAB1787 | 19 | 6 | (1) *Acyrthosiphon pisum* |
| AAB2572 | 16 | 5 | (2) *Aulacorthum solani, Macrosiphum gei* |
| AAB4239 | 20 | 4 | (3) *Macrosiphum rosae, M. funestum, Sitobion rosivorum* |
| AAB4894 | 17 | 5 | (1) *Sitobion avenae* |
| AAB6874 | 10 | 5 | (6) *Ericaphis scammelli, E. fimbriata, Rhodobium porosum, Wahlgreniella nervata, W. vaccinii, W. arbuti* |
| AAB7937 | 30 | 6 | (8) *Aphis craccivora, A. masoni, A. intybi, A. rumicis, A. spiraecola, A. tirucallis, A. coronillae, A. fabae* |
| AAB8566 | 20 | 5 | (2) *Hyperomyzus lactucae, H. carduellinus* |
| AAB9726 | 14 | 6 | (1) *Brachycaudus helichrysi* |
| AAC1165 | 13 | 5 | (2) *Brachycaudus cardui, B. lateralis* |
| AAC1372 | 22 | 6 | (1) *Aphis nerii* |
| AAC1374 | 8 | 3 | (5) *Aphis nasturtii, A. davletshinae, A. umbrella, A. althaeae, A. cf. rostella* |
| AAD0145 | 18 | 6 | (1) *Brevicoryne brassicae* |
| AAD0902 | 4 | 3 | (1) *Nearctaphis bakeri* |
| AAD4538 | 12 | 6 | (1) *Rhopalosiphum maidis* |
| AAD9153 | 11 | 6 | (2) *Lipaphis pseudobrassicae, L. erysimi* |
| AAE2497 | 13 | 5 | (1) *Hysteroneura setariae* |
| AAG3896 | 14 | 5 | (1) *Tetraneura nigriabdominalis* |
| AAG6658 | 6 | 3 | (1) *Chromaphis juglandicola* |
| AAH2863 | 4 | 3 | (1) *Periphyllus lyropictus* |
| AAI0406 | 13 | 5 | (1) *Rhopalosiphum rufiabdominale* |
| AAI4332 | 3 | 2 | (NA) Identified to genus–*Aphis* |
| AAI7650 | 2 | 1 | (1) *Baizongia pistaciae* |
| AAK5331 | 3 | 2 | (1) *Hyadaphis coriandri* |
| AAK7235 | 22 | 5 | (2) *Melanaphis sacchari, M. japonica* |
| AAM0964 | 7 | 4 | (2) *Macrosiphoniella yomogifoliae, M. abrotani* |
| AAN2425 | 5 | 4 | (1) *Macrosiphoniella sanborni* |
| AAN4898 | 2 | 2 | (1) *Brachyunguis harmalae* |
| AAO7083 | 6 | 3 | (1) *Cinara tujafilina* |
| AAP9276 | 5 | 3 | (NA) Identified to genus–*Schizaphis* |
| AAX9332 | 4 | 3 | (1) *Sarucallis kahawaluokalani* |
| AAY6004 | 4 | 2 | (1) *Melanaphis donacis* |
| ACD8115 | 2 | 1 | (1) *Hyalopterus pruni* |
| ACF2924 | 11 | 6 | (1) *Rhopalosiphum padi* |
| ACI9922 | 5 | 3 | (NA) Identified to genus–*Capitophorus* |
| ACO4203 | 4 | 2 | (1) *Schizaphis rotundiventris* |
| ACO5373 | 2 | 1 | (1) *Hyalopterus pruni* |
| ACS1400 | 2 | 1 | (1) *Acyrthosiphon gossypii* |

*(Continued)*

**Table 3.** (Continued)

| BIN | Countries | Continents | (Number) and names of the associated species |
|---|---|---|---|
| ABY0239 | 1 | 1 | (NA) Identified to genus–*Aphis* |
| ACP3887 | 1 | 1 | (NA) Identified to genus–*Forda* |
| ACS1208 | 1 | 1 | (NA) Identified to genus–*Acyrthosiphon* |
| ACS1445 | 1 | 1 | (1) *Acyrthosiphon malvae* |
| ACS2175 | 1 | 1 | (NA) Identified to subfamily–*Aphidinae* |
| ACT3010 | 1 | 1 | (NA) Identified to subfamily–*Aphidinae* |
| ACV1458 | 1 | 1 | (1) *Uroleucon sonchi* |
| ACV6041 | 1 | 1 | (NA) Identified to genus–*Aphis* |

Prior studies revealed a strong correspondence between BINs and known species [39], in particular when reference specimens are identified by experts [84]. The same pattern was apparent in this study as 38 of 41 species were assigned to a single BIN. There were only two exceptions; *R. padi* was assigned to two BINs and *A. gossypii*–*A. affinis* were assigned to the same BIN. By comparison, when barcode sequences from conspecific specimens from other countries were considered, 12 of the 42 species showed a BIN split, an outcome which likely indicates incorrectly identified specimens [39]. Interestingly, the BIN (AAA3070) shared by specimens of *A. gossypii* and *A. affinis* from Pakistan included 31 additional species names when all records for it on BOLD were considered. Misidentifications and overlooked cryptic species may often cause conflicts between BIN and species morphology [85], but this can only be resolved by detailed taxonomic studies [86]. As well, heteroplasmy, hybridization, and incomplete lineage sorting can also cause BIN-morphology conflicts [87,88]. Furthermore, host affinities of sympatric populations, which have been observed in aphids, also expand intraspecific divergence [89], possibly resulting in BIN splits as we observed in *R. padi*.

Geo-distance correlations showed that the genetic divergence increased with geographic distance in almost half of the aphid species. Interestingly, the inclusion of conspecific sequences from other regions also increased the incidence of BIN splits. Since these analyses included all the conspecific sequences on BOLD, this outcome may reflect taxonomic errors [90]. Although spatial variation in conspecific sequences sometimes leads to increased intraspecific divergence values [91], it is usually too low to reduce the capacity of DNA barcodes to deliver reliable species identifications [47,92].

BINs are valuable in evaluating the geographic range of aphid species because they circumvent taxonomic uncertainties. In addition, BINs are gaining increased use to estimate species numbers [41] and to understand their distributions [52]. This analysis revealed that 27 of the 44 BINs with prior records on BOLD occurred on four or more continents, highlighting the broad ranges of many pest aphids. For example, BINs for *Aphis fabae* (black bean aphid), *A. nerii* (oleander aphid), *A. craccivora* (groundnut aphid), *Acyrthosiphon pisum* (pea aphid), *Brachycaudus helichrysi* (plum aphid), *Brevicoryne brassicae* (cabbage aphid), *L. pseudobrassicae* (turnip aphid), *R. padi* (oat aphid), *R. maidis* (corn aphid), *Macrosiphum euphorbiae* (potato aphid), *M. persicae* (peach aphid), and *Therioaphis trifolii* (alfalfa aphid) were all recorded from six continents. Interestingly, BINs associated with some of these species were also linked with other species on BOLD. For instance, AAA3070 was linked to 33 other species of *Aphis* while AAA6213 was associated with 13 species of *Macrosiphum*, and AAA5565 with nine species of *Aphis*. Although some of these cases may involve BIN sharing by different species [29], most cases likely reflect misidentifications.

The level of BIN overlap between the aphid fauna of Pakistan is much higher (85%) than levels for moths (44%) [93] and spiders (24%) [94]. This difference, may, be due, in part, to the

fact that the winged alates of aphids can disperse long distances and their dispersal capacity with the broad availability of the crop plants that they attack [95]. Consequently, the number of aphid species known from Europe has increased by 20% in the last 30 years [96] reflecting their transport on produced fruits [52], coupled with shifting environmental regimes. Reports suggest that with every 1°C increase, some 15 additional aphid species were recorded in Europe [97]. In North America, about 18% of all aphid species are introduced, and nearly half are plant pests [98]. Rapid developments in DNA sequencing are enabling the documentation of pest species and their distribution across the globe, but conflicts between taxonomic assignments and sequences have limited the full utility of these data. Given this difficulty, the BIN system provides an alternative path to document and track the pest species on a planetary scale.

## Supporting information

**S1 Table. Plant-host family range for aphid species/BINs collected in Pakistan.**
(XLSX)

**S2 Table. Identification and BIN assignment of 809 specimens of Aphididae collected from 123 plant hosts in Pakistan.**
(XLSX)

## Acknowledgments

We are very grateful to C Favret, University of Montreal, Quebec, Canada, for his help in aphid identification. We thank staff at the CCDB for aiding the sequence analysis which was made possible by a grant from Genome Canada and Ontario Genomics in support of the International Barcode of Life (iBOL) project. This study was also enabled by support from the Canada First Research Excellence Fund provided to the Food From Thought research program.

## Author Contributions

**Conceptualization:** Muhammad Tayyib Naseem, Muhammad Ashfaq, Paul D. N. Hebert.

**Data curation:** Muhammad Tayyib Naseem, Muhammad Ashfaq, Paul D. N. Hebert.

**Formal analysis:** Muhammad Tayyib Naseem, Muhammad Ashfaq.

**Funding acquisition:** Muhammad Ashfaq, Paul D. N. Hebert.

**Investigation:** Muhammad Tayyib Naseem, Muhammad Ashfaq, Akhtar Rasool, Paul D. N. Hebert.

**Methodology:** Muhammad Tayyib Naseem, Muhammad Ashfaq, Arif Muhammad Khan, Akhtar Rasool.

**Project administration:** Muhammad Ashfaq, Muhammad Asif, Paul D. N. Hebert.

**Resources:** Muhammad Ashfaq, Paul D. N. Hebert.

**Software:** Muhammad Tayyib Naseem, Muhammad Ashfaq, Arif Muhammad Khan.

**Supervision:** Muhammad Ashfaq, Muhammad Asif, Paul D. N. Hebert.

**Validation:** Muhammad Tayyib Naseem, Muhammad Ashfaq, Paul D. N. Hebert.

**Visualization:** Muhammad Tayyib Naseem.

**Writing – original draft:** Muhammad Tayyib Naseem, Arif Muhammad Khan, Akhtar Rasool, Muhammad Asif.

**Writing – review & editing:** Muhammad Ashfaq, Paul D. N. Hebert.

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
