## [Decision Letter · Decision Letter 0]

23 Oct 2019

PONE-D-19-19598

BIN overlap confirms transcontinental distribution of pest aphids (Hemiptera: Aphididae)

PLOS ONE

Dear Muhammad Ashfaq,

Thank you for submitting your manuscript to PLOS ONE. After careful consideration, we feel that it has merit but does not fully meet PLOS ONE’s publication criteria as it currently stands. Therefore, we invite you to submit a revised version of the manuscript that addresses the points raised during the review process.

We would appreciate receiving your revised manuscript by Dec. 21th. To enhance the reproducibility of your results, we recommend that if applicable you deposit your laboratory protocols in protocols.io, where a protocol can be assigned its own identifier (DOI) such that it can be cited independently in the future. For instructions see: http://journals.plos.org/plosone/s/submission-guidelines#loc-laboratory-protocols

We look forward to receiving your revised manuscript.

Kind regards,

Feng ZHANG, Ph.D.

Academic Editor

PLOS ONE

Journal Requirements:

Additional Editor Comments (if provided):

Reviewers' comments:

Reviewer's Responses to Questions

**Comments to the Author**

1. Is the manuscript technically sound, and do the data support the conclusions?

Reviewer #1: Partly

Reviewer #2: Yes

Reviewer #3: Yes

2. Has the statistical analysis been performed appropriately and rigorously? 

Reviewer #1: Yes

Reviewer #2: Yes

Reviewer #3: Yes

3. Have the authors made all data underlying the findings in their manuscript fully available?

Reviewer #1: Yes

Reviewer #2: Yes

Reviewer #3: Yes

4. Is the manuscript presented in an intelligible fashion and written in standard English?

Reviewer #1: Yes

Reviewer #2: Yes

Reviewer #3: Yes

5. Review Comments to the Author

Reviewer #1: Line1-2. Basically the manuscript reports a barcode reference library for some Pakistan aphid species, I don't think it's appropriate to use a title focusing on distribution of pest aphids. Even without a DNA barcoding study, I still can know the geographic distribution patterns of species.

Line19-38. The abstract with many numbers may let readers get confused. The authors may keep main results and numbers only.

Line43, Line44, Line293. The Aphid Species File is a dynamically updating taxonomic reference, so it's not appropriate to cite a version in 2017. And therefore authors need to check the number of species in related groups.

Line165-179. For a DNA barcoding research, the morphologically identification maybe the first step and detail statistics should be placed better in Methods section.

Line215. 'across much of Pakistan', based on the sampling map, I think "much of Pakistan" is inappropriate.

Line225-230. The discussion here may have logic problem. Specimens could only be identified to a genus and especially subfamily levels can be only due to the authors are not familiar with some aphid groups. If you say the reason is about 'cryptic species complexes', meaning at least the samples can be identified genus level based on sequences. And for the cotton aphid example, the authors cited two relatively old references, do you think the information of 'at least 20 morphologically indistinguishable species' is OK now?

One lack of the current version is that, for their own dataset, the authors discuss very little about the 'conflicts between taxonomic assignments and sequences' or 'BIN split' and 'BIN merger' based on biology of related aphids. So the depth of current discussion is not enough.

Another problem is, whether the authors did think about the sequence quality in the BOLD system? Sequence with problem may lead to wrong split or merger. The fact is that BOLD sequences have quality problem for many insect groups. At least, the authors need to mention and discuss their opinion about this.

Reviewer #2: The manuscript “BIN overlap confirms transcontinental distribution of pest aphids” (PONE-D-19-19598) by Naseem and co-authors analyzes 809 DNA barcodes of aphids from Pakistan using the BOLD workbench. Furthermore, they combined their sequence data with already available DNA barcodes, revealing a broad distribution of pest aphids across six continents.

It is obvious that DNA barcoding as well as upcoming metabarcoding approaches using high throughput sequencing technologies will play a more and more important role in order to document and assess biodiversity in the near future. Therefore we need more comprehensive sequence libraries for a correct identification. The topic of this manuscript is interesting and appropriate for PLoS ONE, indicating a relevant application of DNA barcoding in order to identify an economically and ecologically important group of species. However, various concerns remain and have to be corrected before the manuscript is suitable for publication.

High-throughput sequencing technologies will play an essential role in modern biodiversity research. However, no aspects of this topic are discussed so far. Furthermore, I miss a discussion of possible pitfalls of using DNA barcodes and mtDNA in general in terms of specimen identification, e.g., the existence of incomplete lineage sorting, heteroplasmy etc. as part of the introduction or discussion.

Please find some specific comments made via sticky notes on the PDF file of the manuscript.

Reviewer #3: This paper provides valuable data of aphid DNA barcodes. The sampling is limited to Pakistan, but focusses on pest species, increasing the applicative value of the data, especially as most pests have wide distributions. The paper does not contain much scientific novelty besides adding new valuable records to the barcode databases. The paper nicely demonstrates the benefits of BINs in assigning specimens into the same taxonomic units when this is otherwise hampered by misidentifications and taxonomic uncertainties. I recommend the publication after a very minor review. I do not have any major comments, although I find that in some places the authors are close of drawing too strong conclusions based on assumptions. For example, they conclude some issues being because of misidentifications in other works, although they hardly could show this being certainly the case.

I recommend carefully checking the sentences at lines 270-274 as the sentences there do not appear fully clear to me. For example: “…are they use widely crop plants…” (perhaps “as they use…”). Similarly. “…their transport on produce fruits…”” (perhaps: “…their transport on produced fruits…”).

It is my odd hobby to spot glitches in the references. Here we go:

References 24 and 25: One initial perhaps lacking with the author Hebert

Reference 48. Kirichenkoa > Kirichenko

Lines 458-471: line spacing deviates from that of other parts

Reference 89: EXploitation > Expointation (supposedly)

Reference 91: Footit > Foottit

Disclaimer: I am neither Mr Hebert, nor Mrs Kirichenko, nor Mr Foottit.

Additional small glitches:

Line 559: The word “Linnean” has a different font size.

Figure 3: The name “Sarucallis kahawaluokalani” is not italicized

Figure 3: The name “Aphidinae” should not be italicized as being a subfamily-level name.

6. PLOS authors have the option to publish the peer review history of their article (what does this mean?). If published, this will include your full peer review and any attached files.

Reviewer #1: No

Reviewer #2: No

Reviewer #3: No

---

## [Author Response · Author response to Decision Letter 0]

4 Nov 2019

November 4, 2019

PONE-D-19-19598: BIN overlap confirms transcontinental distribution of pest aphids (Hemiptera: Aphididae)

Response to Reviewers

The comments from the Reviewers are included in italics followed by our response.

Reviewer Comments to Author:

Reviewer #1:

COMMENT: Line1-2. Basically the manuscript reports a barcode reference library for some Pakistan aphid species, I don't think it's appropriate to use a title focusing on distribution of pest aphids. Even without a DNA barcoding study, I still can know the geographic distribution patterns of species.

Response: The use of BIN assignments to map the geographic distribution of aphids and other species of pest insects is a key message of our study and this is reflected in the title. We certainly agree with the reviewer that species distributions can be mapped in other ways (e.g. via morphological studies by taxonomists). However, since the taxonomic expertise to correctly identify species is often not available, additional/alternative tools are needed to support species identification. DNA barcoding/BINs meet this need. The utility of the BINs system is further enhanced by the capacity of high-throughput sequencers to support metabarcoding, an approach which allows the species composition of bulk samples to be rapidly assessed. 

COMMENT: Line19-38. The abstract with many numbers may let readers get confused. The authors may keep main results and numbers only.

Response: Several of the less important numbers were removed from the Abstract. 

COMMENT: Line43, Line44, Line293. The Aphid Species File is a dynamically updating taxonomic reference, so it's not appropriate to cite a version in 2017. And therefore authors need to check the number of species in related groups.

Response: Reference [3] has been updated to the format suggested by the database developer. 

COMMENT: Line165-179. For a DNA barcoding research, the morphologically identification maybe the first step and detail statistics should be placed better in Methods section.

Response: Detailed statistics were included in the Results section because DNA barcoding was used to support and facilitate the morphological identifications and finalize the species names. We have mentioned this in the first line of the Results. 

COMMENT: Line215. 'across much of Pakistan', based on the sampling map, I think "much of Pakistan" is inappropriate.

Response: “across much of Pakistan” has been changed to “major agricultural areas of Pakistan” since the areas covered in this study are mostly agricultural lands and plains. 

COMMENT: Line225-230. The discussion here may have logic problem. Specimens could only be identified to a genus and especially subfamily levels can be only due to the authors are not familiar with some aphid groups. If you say the reason is about 'cryptic species complexes', meaning at least the samples can be identified genus level based on sequences. And for the cotton aphid example, the authors cited two relatively old references, do you think the information of 'at least 20 morphologically indistinguishable species' is OK now?

Response: We agree with the reviewer’s comment and have expanded the statement by adding – “or their identification was beyond our expertise”. 

About cotton aphid (Aphis gossypii): Citations supporting the morphological complexity of A. gossypii are from 2007 (Blackman & Eastop) and 2014 (Favret). These authors are renowned aphid taxonomists and their conclusions on A. gossypii have not been challenged. 

Based on a more recent report (2014), the statement on the number of species in A. gossypii group has been revised and the old citation [76; Stroyan 1984] has been replaced with a new one (Lagos-Kutz et al. 2014). 

An old citation on the range of host plants [77, 1955] has been replaced with a more recent report by Singh G, Singh NP, Singh R, Singh G, Singh NP (2014). Food plants of a major agricultural pest Aphis gossypii Glover (Homoptera: Aphididae) from India: An updated checklist. International Journal of Life Sciences Biotechnology and Pharma Research. 2014;3: 1–26.

COMMENT: One lack of the current version is that, for their own dataset, the authors discuss very little about the 'conflicts between taxonomic assignments and sequences' or 'BIN split' and 'BIN merger' based on biology of related aphids. So the depth of current discussion is not enough.

Response: One species in our dataset was assigned to two BINs (BIN split) and two shared a BIN (BIN merger). These cases of BIN-morphology conflicts (split and merger) have now been discussed (lines 377-380; revised manuscript with track changes). 

COMMENT: Another problem is, whether the authors did think about the sequence quality in the BOLD system? Sequence with problem may lead to wrong split or merger. The fact is that BOLD sequences have quality problem for many insect groups. At least, the authors need to mention and discuss their opinion about this.

Response: BOLD follows stringent quality criteria and sequences not meeting them (<1% ambiguous bases, no stop codon or contamination flag) are not assigned a BIN. This has already been stated in the Data Analysis section of the Methods. Only high quality, validated sequences were included in the current study. 

BOLD is a public resource and allows users to submit sequences and specimen data without prior permission. However, it does provide quality filters that allow users to exclude invalid and poor quality sequences from any analysis. 

Reviewer #2: 

COMMENT: The manuscript “BIN overlap confirms transcontinental distribution of pest aphids” (PONE-D-19-19598) by Naseem and co-authors analyzes 809 DNA barcodes of aphids from Pakistan using the BOLD workbench. Furthermore, they combined their sequence data with already available DNA barcodes, revealing a broad distribution of pest aphids across six continents.

It is obvious that DNA barcoding as well as upcoming metabarcoding approaches using high throughput sequencing technologies will play a more and more important role in order to document and assess biodiversity in the near future. Therefore we need more comprehensive sequence libraries for a correct identification. The topic of this manuscript is interesting and appropriate for PLoS ONE, indicating a relevant application of DNA barcoding in order to identify an economically and ecologically important group of species. However, various concerns remain and have to be corrected before the manuscript is suitable for publication.

High-throughput sequencing technologies will play an essential role in modern biodiversity research. However, no aspects of this topic are discussed so far. Furthermore, I miss a discussion of possible pitfalls of using DNA barcodes and mtDNA in general in terms of specimen identification, e.g., the existence of incomplete lineage sorting, heteroplasmy etc. as part of the introduction or discussion.

Response: We have added a statement on the integration of DNA barcoding with high-throughput sequencing workflows to the Introduction. A discussion point on the potential shortcomings of DNA barcoding (heteroplasmy, incomplete lineage sorting) has been included in the Discussion. 

Please find some specific comments made via sticky notes on the PDF file of the manuscript.

COMMENT: L28: Why 801 and not 809 (see above)?

Response: This difference reflects the fact that only sequences meeting quality standards (>500 bp, <1% ambiguous bases, no stop codon or contamination flag) were assigned a BIN. Eight sequences didn’t meet the BIN standard. This has been clarified on lines 128-129 in the Methods section. 

COMMENT: L41: It would be nice to see some images of the analyzed species. Use the options that open access and PLOS ONE offer!

Response: The microscope used for morphological identification of aphids lacked a high quality camera. As a result, most of the images lack the resolution to make morphological characters useful for species assignments. However, as mentioned in the manuscript (line 106), all images taken for the study are accessible on BOLD. 

COMMENT: L45: Sounds a little bit to drastic; please change.

Response: “Attack” was changed to “damage”. 

COMMENT: L52: Cybertaxonomy per se does not represent an "alternative approach for identification". Instead of this, it focusses on comprehensive descriptions allowing valid identifications (e.g., Organisms, Diversity and Evolution 16: 1-12). Please change this part. 

Response: Cybertaxonomy and the related reference have been removed. A new reference on the use of protein profiling for species identification has been added.

Jayaseelan M, Roesler Uwe R. MALDI-TOF MS Profiling-Advances in species identification of pests, parasites, and vectors. Front Cell Infect Microbiol. 2017;7: 184.

COMMENT: L70: Keep in mind that BIN assignments on BOLD are constant¬ly updated as new sequences are added, splitting and/or merging individual BINs in light of new data. Therefore, BINs are not written in stone and can change. Therefore it is important to document the specific date of BIN assignment. 

Response: The date of the BIN assignment has been documented in the Methods section (line 131).

COMMENT: L76: See also PLoS ONE 9(9) e106940.

Response: Suggested reference (PLoS ONE 9(9) e106940) has been cited. 

COMMENT: L113: 94 °C (a gap between the number and °C here and in the following).

Response: Corrected. 

COMMENT: L123: What about the deposition of vouchers?

Response: Information on the deposition of vouchers has been added (line 126-128).

COMMENT: L134: Think about updating your MEGA-version. The most recent version is 10!

Response: We thank the Reviewer for this suggestion. We used MEGA v6 just because it has been well tested. We will consider the newer version for our next study. 

COMMENT: L223: Please rephrase

Response: Sentence has been rephrased to clarify the statement. 

Reviewer #3: 

COMMENT: This paper provides valuable data of aphid DNA barcodes. The sampling is limited to Pakistan, but focusses on pest species, increasing the applicative value of the data, especially as most pests have wide distributions. The paper does not contain much scientific novelty besides adding new valuable records to the barcode databases. The paper nicely demonstrates the benefits of BINs in assigning specimens into the same taxonomic units when this is otherwise hampered by misidentifications and taxonomic uncertainties. I recommend the publication after a very minor review. I do not have any major comments, although I find that in some places the authors are close of drawing too strong conclusions based on assumptions. For example, they conclude some issues being because of misidentifications in other works, although they hardly could show this being certainly the case.

Response: We thank the reviewer for these valuable comments and suggestions. Although our sampling program was limited to one nation (Pakistan), we were able to extend the analysis to global scale by exploiting the BIN system. Our statements on issues related to misidentification of species are not meant to diminish the value of morphological identification. We simply highlight the difficulties of gaining reliable morphological identifications in areas where the required taxonomic specialists are lacking. 

COMMENT: I recommend carefully checking the sentences at lines 270-274 as the sentences there do not appear fully clear to me. For example: “…are they use widely crop plants…” (perhaps “as they use…”). Similarly. “…their transport on produce fruits…”” (perhaps: “…their transport on produced fruits…”).

Response: These sentences have been corrected. 

COMMENT: It is my odd hobby to spot glitches in the references. Here we go:

References 24 and 25: One initial perhaps lacking with the author Hebert

Reference 48. Kirichenkoa > Kirichenko

Lines 458-471: line spacing deviates from that of other parts

Reference 89: EXploitation > Expointation (supposedly)

Reference 91: Footit > Foottit

Response: These references and line spacing have been corrected. Exploitation is the original word used by the authors, so it was not changed. 

COMMENT: Additional small glitches:

Line 559: The word “Linnean” has a different font size.

Figure 3: The name “Sarucallis kahawaluokalani” is not italicized

Figure 3: The name “Aphidinae” should not be italicized as being a subfamily-level name.

Response: Corrected.

---

## [Decision Letter · Decision Letter 1]

19 Nov 2019

PONE-D-19-19598R1

BIN overlap confirms transcontinental distribution of pest aphids (Hemiptera: Aphididae)

PLOS ONE

Dear Muhammad,

Thank you for submitting your manuscript to PLOS ONE. After careful consideration, we feel that it has merit but does not fully meet PLOS ONE’s publication criteria as it currently stands. Therefore, we invite you to submit a revised version of the manuscript that addresses the points raised during the review process.

We would appreciate receiving your revised manuscript by Dec. 19th, 2019. To enhance the reproducibility of your results, we recommend that if applicable you deposit your laboratory protocols in protocols.io, where a protocol can be assigned its own identifier (DOI) such that it can be cited independently in the future. For instructions see: http://journals.plos.org/plosone/s/submission-guidelines#loc-laboratory-protocols

We look forward to receiving your revised manuscript.

Kind regards,

Feng ZHANG, Ph.D.

Academic Editor

PLOS ONE

Reviewers' comments:

Reviewer's Responses to Questions

**Comments to the Author**

1. If the authors have adequately addressed your comments raised in a previous round of review and you feel that this manuscript is now acceptable for publication, you may indicate that here to bypass the “Comments to the Author” section, enter your conflict of interest statement in the “Confidential to Editor” section, and submit your "Accept" recommendation.

Reviewer #2: (No Response)

Reviewer #3: All comments have been addressed

2. Is the manuscript technically sound, and do the data support the conclusions?

Reviewer #2: Yes

Reviewer #3: Yes

3. Has the statistical analysis been performed appropriately and rigorously? 

Reviewer #2: Yes

Reviewer #3: Yes

4. Have the authors made all data underlying the findings in their manuscript fully available?

Reviewer #2: Yes

Reviewer #3: Yes

5. Is the manuscript presented in an intelligible fashion and written in standard English?

Reviewer #2: Yes

Reviewer #3: Yes

6. Review Comments to the Author

Reviewer #2: It is nice to see that the authors accepted most comments and/or remarks. Some minor concerns still remain (see sticky notes). After correction, the manuscript should be suitable for publication in PLoS ONE.

Reviewer #3: I like that the authors have carefully addressed both the issues that I myself spotted, but also the issues spotted by the other reviewers. I noticed that one of my own comments was with a mistake. I definitely did not mean to change the word "exploitation" to "expointation" in the reference list, but to remove the double capital in the previous word (EXploitation > Exploitation).

7. PLOS authors have the option to publish the peer review history of their article (what does this mean?). If published, this will include your full peer review and any attached files.

Reviewer #2: No

Reviewer #3: No

---

## [Author Response · Author response to Decision Letter 1]

19 Nov 2019

November 19, 2019

PONE-D-19-19598: BIN overlap confirms transcontinental distribution of pest aphids (Hemiptera: Aphididae).

Response to Reviewers

The comments from the Reviewers are included in italics followed by our response.

Reviewer Comments to Author:

Reviewer #2: 

COMMENT: It is nice to see that the authors accepted most comments and/or remarks. Some minor concerns still remain (see sticky notes). After correction, the manuscript should be suitable for publication in PLoS ONE.

Response: All the comments, pointed out through sticky notes on the pdf, have been addressed in the revised version. 

COMMENT: L137: Which program was used? MUSCLE? You should mention it (inc. giving the reference).

Response: Nucleotide sequence alignment was performed using ClustalW. This has been mentioned, with relevant reference, in the revised version. 

COMMENT: L299: Most DOIs and PMIDs are still missing

Response: We have followed the reference style for PLOS ONE. This style does not ask for DOIs and PMIDs when the volume and page numbers are available.

Reviewer #3: 

COMMENT: I like that the authors have carefully addressed both the issues that I myself spotted, but also the issues spotted by the other reviewers. I noticed that one of my own comments was with a mistake. I definitely did not mean to change the word "exploitation" to "expointation" in the reference list, but to remove the double capital in the previous word (EXploitation > Exploitation).

Response: The comment has been addressed.

---

## [Editor Report · Decision Letter 2]

25 Nov 2019

BIN overlap confirms transcontinental distribution of pest aphids (Hemiptera: Aphididae)

PONE-D-19-19598R2

Dear Dr. Muhammad Ashfaq,

We are pleased to inform you that your manuscript has been judged scientifically suitable for publication and will be formally accepted for publication once it complies with all outstanding technical requirements.

With kind regards,

Feng ZHANG, Ph.D.

Academic Editor

PLOS ONE
---

## [Editor Report · Acceptance letter]

2 Dec 2019

PONE-D-19-19598R2 

BIN overlap confirms transcontinental distribution of pest aphids (Hemiptera: Aphididae) 

Dear Dr. Ashfaq:

I am pleased to inform you that your manuscript has been deemed suitable for publication in PLOS ONE. Congratulations! Your manuscript is now with our production department. 

With kind regards,

on behalf of

Dr. Feng ZHANG 

Academic Editor

PLOS ONE